# Prediction of the Compressive Strength of Waste-Based Concretes Using Artificial Neural Network

**DOI:** 10.3390/ma15207045

**Published:** 2022-10-11

**Authors:** Mouhamadou Amar, Mahfoud Benzerzour, Rachid Zentar, Nor-Edine Abriak

**Affiliations:** 1IMT Nord Europe, Institut Mines-Télécom, Centre for Materials and Processes, F-59000 Lille, France; 2Univ. Lille, Institut Mines-Télécom, Univ. Artois, Junia, ULR 4515—LGCgE—Laboratoire de Génie Civil et géoEnvironnement, F-59000 Lille, France

**Keywords:** artificial neural network, concrete, mineral additions, prediction, formulation, compressive strength

## Abstract

In the 21st century, numerous numerical calculation techniques have been discovered and used in several fields of science and technology. The purpose of this study was to use an artificial neural network (ANN) to forecast the compressive strength of waste-based concretes. The specimens studied include different kinds of mineral additions: metakaolin, silica fume, fly ash, limestone filler, marble waste, recycled aggregates, and ground granulated blast furnace slag. This method is based on the experimental results available for 1303 different mixtures gathered from 22 bibliographic sources for the ANN learning process. Based on a multilayer feedforward neural network model, the data were arranged and prepared to train and test the model. The model consists of 18 inputs following the type of cement, water content, water to binder ratio, replacement ratio, the quantity of superplasticizer, etc. The ANN model was built and applied with MATLAB software using the neural network module. According to the results forecast by the proposed neural network model, the ANN shows a strong capacity for predicting the compressive strength of concrete and is particularly precise with satisfactory accuracy (R² = 0.9888, MAPE = 2.87%).

## 1. Introduction

According to the World Bank, the amount of global solid waste is currently 2.2 billion tons per year. This figure is likely to increase due to global demographic and economic growth. The overconsumption and inefficient use of materials also have a critical impact on the environment and climate. The annual cement needs in France are 16.9 million tons (in 2020), and in 2020 the need for aggregates was over 400 million tons, 96% of natural origin. Concrete is an ancient and widely used material because of its mechanical properties, which have long been appreciated. The Egyptians were already using it around 2600 BC in the pyramid of Abu Rawash. This material continued to evolve until the invention of reinforced concrete in 1867 by Joseph Monier (1823–1906). Concrete is a multiphase material consisting of a granular skeleton and a cementitious matrix. Due to chemical reactions of hydration followed by hardening, the assembly stiffens. This attributes some specific physicochemical properties to the material: elastic properties, high compressive strength, low tensile strength, permeability, etc. These properties are the result of a series of chemical reactions, triggered as soon as the anhydrous cement and water come into contact. These reactions thus give rise initially to portlandite (Ca(OH)_2_), which acts as a trigger for setting, then to various hydrates (C-S-H, C-A-H, C-A-S-H, etc.) representing the “glue” of the matrix.

Concrete structures now represent more than 90% of modern structures. Concrete has thus become the most important building material on the planet, in terms of volume and turnover. Its success stems from, among other things, its extraordinary versatility and availability virtually anywhere on Earth, and its durability. It is at the dawn of the 21st century that mankind is confronted with an unprecedented paradigm: How to ensure the sustainability of nature and biodiversity for future generations and at the same time meet current growing economic needs: energy, materials, and resources of all kinds?

The construction sector is one of the most important in Europe. From an environmental standpoint, it represents 30% of carbon dioxide emissions (2009). Cement production is an important source of CO_2_ (5 to 7% of worldwide emissions). The concrete industry worldwide consumes annually over 8–12 billion tons of natural aggregates [1]. This is why the use of binders based on additional byproduct materials, such as metakaolin, fly ash, sludge ash, blast-furnace slag, or silica fume, is gaining more and more interest and their efficiency has been proven [2,3,4]. Because of the environmental issues explained above and because they are a way to enhance the durability of structures exposed to harsh environments, their utility is fundamental [2].Several formulations and prediction models (Feret, Bolomey, Abram, Powers [4,5,6,7]) have proposed that the compressive strength R_c_ of concrete depends mainly on:i.The grade of the cement, the age, and the method of curing;ii.The water to binder ratio, as well as the compactness of the granular skeleton.

Currently, there is a lack of research on comprehensive methods for predicting the strength of byproduct-based concrete or mortar. Hence, authors usually use correlation or some empirical or non-adapted formulae.

Recently, numerous studies have forecast the compressive strength of cementitious materials using extrapolation methods, compressible packing models, regression analysis methods, genetic algorithms, fuzzy logic, and artificial neural networks (ANNs) [8,9]. However, among these approaches, the ANN seems a relevant and efficient method due to its ability to learn from input and output relationships in complex problems [10]. Moreover, the ANN is suitable for modeling different properties of concrete, for mapping its mechanical characteristics, including compressive and tensile strength, slump, filling capacity, and segregation, and for many types of concrete [8,9,10].

In recent years, several studies have reported that ANNs can be used to solve engineering problems. However, the required data may be complex or insufficient [11] for estimating the compressive strength of concretes [12,13]. These studies involved issues related to high performance [1,14,15,16,17,18,19], self-compacting concrete [20,21,22,23], and lightweight concretes [24,25,26], sulphate resistance in concrete [27,28], cyclic behavior of concretes [29], recycled aggregate [1,30,31] and waste material [27,32,33]. Machine learning and AI are steadily gaining interest and over 10,152 papers were published in 2021 in the material engineering field alone.

Faridehmihr et al. [34] explored the use of waste materials, including fly ash (FA), palm oil fly ash (POFA), waste ceramic powder (WCP), and granulated blast-furnace slag (GBFS) in alkali-activated materials. Properties such as the mechanical resistance can be properly predicted using an artificial neural network (ANN) combined with a metaheuristic Krill Herd algorithm (KHA) model. Faridehmihr et al. [34] showed that ANNs are efficient in investigating the cradle-to-gate life-cycle assessment (LCA) of ternary blended alkali-activated mortars. Mhaya [35] evaluated the performance of several modified rubberized concretes by exposing them to aggressive environments. The final mechanical properties were predicted using an ANN combined with particle swarm optimization (PSO). A similar work conducted by Golafshani [36] added to an ANN the use of a multi-objective multi-verse optimizer (MOMVO). Alabduljabbar et al. [37] adopted the same method as that of Sadowski et al. [38] and used an optimized ANN to estimate the mechanical properties in a wide experimental study on the sustainability of employing waste sawdust and supplementary cementitious material (SCM) to make high-performance cement-free lightweight concretes. Ray et al. [39] monitored the consequences of the incorporation of fine glass aggregate and condensed milk can fiber (Sn) on the compressive and splitting tensile strength at three curing ages using an ANN. The results showed very good accuracy. It is clear that the prediction of concrete properties can be efficiently performed using machine learning technology [40,41].

In the early 2000s, several studies [42,43] showed the great potential to optimize mix proportioning and forecasting of concrete properties. Apostolopoulou et al. [44] investigated the use of ANNs to simulate the characteristics of lime-based mortars, such as compressive and flexural strength and consistency. The final results showed that the developed ANN models fit satisfactorily with the experimental data. Gupta et al. [45] used an ANN in a recent study since there was no mathematical model for the rapid prediction of mechanical properties of rubberized concrete. The trained network based on data compiled from recent research showed results that predicted compressive strength, modulus of elasticity (static and dynamic), and mass loss. Several other studies on this type of concrete led to similar conclusions, while others focused on predicting the properties of recycled aggregate-based concrete [35,46]. The test data are generally sets of compressive strength, splitting strength, porosity, the permeability coefficient of recycled aggregate, etc. Based on mean squared error (MSE), root mean square error (RMSE), and coefficient of regression (r^2^), the results proved to have a very good fit, as stated by Dantas et al. [47]. It has been demonstrated that ANNs can predict the compressive and tensile strength of concretes containing construction and agricultural wastes [32,48], blast furnace slag [35], and alkali-activated mortars [34,37]. Some authors combined an ANN with other techniques, such as a genetic algorithm (GA) [35], statistics and holistic models [44], the cuckoo search method [49,50], ANFIS models [24,51], fuzzy logic models [52,53], and the Monte Carlo approach [54], to optimize the prediction results. Jiang et al. [53] and Farooq et al. [55] studied the prediction of mechanical properties of self-compacting concretes and high-performance concretes using an ANN on over 1030 datasets. The excellent findings obtained suggested that machine learning processes are quite robust and efficient, becoming indispensable for concrete property prediction. In addition, Asteris et al. [56] developed a methodology that predicts the effects of seismic loads on masonry structures. The authors were able to take into account the weakness, damage, fragility, and general properties of structures. Ray et al. [39], like Sadowski et al. [38], recently showed that ANN techniques are relevant in predicting properties of waste-based concretes and mineral admixtures such as metakaolin, silica fume, dust-based, filler-based, glass waste-based quartz mineral, and fibers. Bui et al. [57] used a whale optimization algorithm (WOA) coupled with a neural network (NN) with over 400 nodes to simulate the 28-day compressive strength of concrete. The results showed that the WOA-NN is reliable and has the highest correlation of 0.8976 when compared to different techniques of modeling. Other recent studies [58,59] conducted on the prediction of concrete’s compressive strength used several methods (Support Vector Regression (SVR), Decision Tree Regression (DTR), Gradient Boosting Regression (GBR), and ANN) for comparative purposes. It was shown that SVR, DTR, and ANN were reliable methods.

## 2. Research Significance

More than 10 billion tons of concrete are currently used worldwide, and in the USA, for example, USD 9.4 billion would be needed to restore the country’s 600,000 bridges. It is therefore important to emphasize that the emergence of innovative processes and techniques in the formulation and composition of concretes is very much needed. The use of supplementary cementitious material may be one of the best solutions for nature and resource preservation. One of the best-established approaches to reducing the impact of cement on the environment is the replacement of clinker with other materials. This method reduces energy consumption and increases production, without any additional industrial installation [60]. These substitutes are generally reactive byproducts from other industries: granulated blast furnace slag (GBFS), a byproduct of the iron industry; and fly ash (FA), generated by electricity production after the burning of coal. Moreover, natural materials such as calcined clays, pozzolans, and limestone fillers have proved suitable for concrete use. Several theoretical methods exist to predict concrete strength: those of Feret (1897), Abrams (1920), Bolomey (1925) [3,6], etc. Depending on the case, certain preponderant parameters, such as the water/binder ratio (W/L), the substitution rate (p(%)), and the maturation time, may influence the final strength of the concrete. However, these existing formulae are not always adapted to the materials cited above. In this work, ANN modeling and results in predicting concrete properties were investigated. The aim was to develop and set up AI-based tools to predict the properties of concretes containing byproducts reused as Appendix A. This research highlights the potential of using an ANN with satisfactory and reliable results in predicting the characteristics of environmentally friendly concretes [38,61]. The novelty of this article relies on the scale of the dataset used and its extensiveness. Contrarily to several studies, this study tested the concomitant set of multiple supplementary cementitious materials using machine learning.

## 3. Artificial Neural Networks

Artificial neural networks (ANNs), which are part of the machine learning process, involve mathematical techniques based on the conception of interconnected layers of nodes [62]. An artificial neural network (ANN) is an artificial intelligence system that focuses on the identification and solving of complex issues and phenomena. A parallel can be drawn with conventional digital computing techniques, yet neural networks have many additional assets. For instance, they use equivalent processing modes and distributed information storage, and also have high accuracy. Furthermore, these methods are very robust when operating following the training process and are flexible to new information and learning [63]. The ANN system is meant to recreate the biological characteristics of the nerve cell structure of the brain.

Usually, an ANN is made up of an input layer of neurons, which includes other layers within it. These neurons predict the process results [10]. The junction of the layers is based on link weights according to Rafiq et al. [64]. As a definition, it can be said that an ANN is a computing system composed of multiple simple units and highly interconnected processing elements. These elements analyze information through the dynamic state response to external inputs. An ANN is skilled in memorizing the characteristics or features of given data and can match or make connections from new data to old with different levels of success [62,65]. The hidden layers (HLs) play the role of connecter or information carrier. The structure then enables the nets to extract a non-linear correlation from the available dataset [24].

There are six main parts in an ANN around a considered neuron nj: inputs (pi), bias (bj), weights (wij), sum function (n)j, activation function (f), and outputs (aj), as displayed in Figure 1. Inputs can be defined as information considered to be decision variables coming from neurons or the external environment. Weights are values that convey the effect of inputs or process elements on each other. Random weight values can be triggered when the process starts. The sum function is an operation that reflects the whole effect of inputs and weights by taking into account a bias value on this process element [13,66] (Equation (1)).
(1)$${\left(n\right)}_{j}=\overset{i=k}{\underset{i=1}{\sum }}w_{\mathrm{ij}}p_{i}+b_{j}$$
where:

i = [1;k] is the number of the ith input neuron

j = [1;m] is the number of the jth output neuron

k = number of units in the ith input vector.

b_j_ = value of bias (referred to as the activation threshold) associated with jth node.

The activation function or transfer function (usually the log-sigmoid function or the hyperbolic tangent [24]) is a function that processes the (n)_j_ value and then determines the corresponding output value according to the formula in Equation (2) [16,67]. It also represents a way to simulate a phenomenon’s reaction using input and output parameters [68].
(2)$${\left(a\right)}_{j}=\;f{\left(n\right)}_{j}=\frac{1}{1+e^{-\alpha {\left(n\right)}_{j\;}}}$$
where ${\left(a\right)}_{j}$ is the output of the jth neuron and α is a constant used to control the slope of the semi-linear region [13], and usually α=1.

### 3.1. Neuron Model (Logsig, Tansig, Purelin)

In an ANN, each input is weighted with an appropriate *w*. The sum of the weighted inputs and the bias forms the input to the transfer function *f*(n)_j_. Multilayer networks often use the log-sigmoid transfer function logsig(n)_j_. The function logsig(n)_j_ generates outputs between 0 and 1 as the neuron’s net input goes from negative to positive infinity. Alternatively, multilayer networks can use the tan-sigmoid transfer function tansig(n)_j_ or purlin. Logsig(n)_j_ appears to be more adapted to the current study as it was found to be more accurate for some predictions [69]. Several algorithms can be implemented in ANN modeling, such as Bayesian regularization, Scaled Conjugate Gradient, Levenberg–Marquardt, one-step secant, and some other combination rules [19]. The most popular ANN method is the feedforward multilayer perceptron (MLP) system. The general scheme of the adopted neural network system is given in Figure 2. Final weight values come at the end of the training process and their final value are defined based on how well the model was trained.

### 3.2. Training Methods

The neural network models applied in this study were developed using the Neural Network Toolbox in MATLAB software. The models were generated with 02 hidden layers and 10 neurons per hidden layer (Table 1). Of the total data, 70% was used for the training process. In our approach, 15% of the remaining data was used for testing and the other 15% for validation. The training process was operated using the Levenberg–Marquardt backpropagation algorithm (LMBPA), similar to Abu Yaman et al. [20] and Kumar et al. [70]. The LMBPA was chosen due to its simplicity of use. It was also shown that one of the most reliable ANN training algorithms is the backpropagation (BP) algorithm, which distributes the network error to arrive at the best fit or minimum error [71,72] and was, accordingly, used in this study.

### 3.3. Feedforward Network

A feedforward neural network was used in this study. This seems to be the most commonly used ANN architecture type. Feedforward networks have all their neurons classified into different layers. All neurons in each of the considered previous layers are connected to the neurons in the next layer. The multilayer architecture considered in this study, also called a multilayer perception [70], is given in Figure 3. There is no reliable method for deciding the number of neural units or layers required for a particular problem. This comes with experience and trials that are necessary to achieve the best network configuration [9].

The structure using multiple layers of neurons creates nonlinear relationships between input and output vectors. The number of layers determines the complexity of the architecture and the forecast precision. When the training process is completed, a positive value of weight signifies that the corresponding feature is directly related to the output. On the other hand, a negative weight implies that the corresponding feature is inversely linked to the output. The more the weight related to a feature, the more the effect of the corresponding feature on the output.

## 4. The Learning and Testing Process

### 4.1. The Backpropagation Algorithm (BPA)

The term ‘backpropagation’ indicates a method in which a correction gradient is calculated for nonlinear multilayer networks [73]. This step is an essential part of the network learning process and is performed by the learning algorithm [64]. To assess the performance of the neural network model, an error measure such as root mean square error (RMS) can be used [24]. The determination of and reduction in the error value or cost function can be performed using the so-called generalized delta rule [11]. In fact, the error (which is the gap between forecast and actual values) is reduced using a backpropagation algorithm [9]. Then, during the BPA process, the neuron weights are subsequently adjusted (Figure 4). According to Oztas et al. [18], the BPA is one of the most famous and most widely used training algorithms [11]. In a multi-layer perceptron (MLP) this method corresponds to a gradient descent technique that minimizes the error or cost of the process.

LMBPA was used in this present study, as implemented in MATLAB and its neural network fitting module. The LMBPA is the fastest backpropagation algorithm for many engineering problems and is highly recommended as a first-choice supervised algorithm, according to Sobhani et al. [10]. However, it requires more memory than other algorithms such as the Momentum, Adagrad, and Rmsprop methods.

A cost function (error function) can be defined to quantify the difference between the actual value and desired (forecast) outputs (Equation (3)):(3)$$J\;\left(w_{ij}\right)=\frac{1}{2}*\left(\sum_{i}\sum_{j}{\left(a_{PREDICT}-a_{TARGET}\right)}_{j}{}^{2}\right)$$
where:

*a_PREDICT_* = *a _(j, PREDICT)_* is the forecast value,

*a_TARGET_* = *a _(j, TARGET)_* is the experimental value.

Gradient descent is an optimization algorithm that approaches a local minimum of a function by taking steps proportional to the negative of the gradient of the function at the current point. The main objective of the algorithm is then to reduce the cost value and adjust the weight that must be updated very smoothly and slowly by iteration until convergence.

In the gradient descent technique, the adjusted weight can be expressed as (Equation (4)):(4)$$\begin{array}{c} w_{ij}\left[n+1\right]=w_{ij}\;\left[n\right]+\tau *\;\frac{▽J\;\left(w_{ij}\right)\;}{▽\left(w_{ij}\right)}\left[n\right]\\ \mathrm{new}\;\mathrm{weight}=\mathrm{old}\;\mathrm{weight}\;-\mathrm{derivative}\;\mathrm{Rate}\;*\;\mathrm{learning}\;\mathrm{rate} \end{array}$$ where τ is known as the step-size parameter and affects the rate of convergence of the algorithm, and $\frac{\nabla J\;\left(w_{ij}\right)}{\nabla \left(w_{ij}\right)}$ is the derivative rate or gradient of the loss function $J\;(w_{ij}$).

The learning process consists of changing the weights in order to minimize this J(w) in a gradient descent technique. The training process is considered as successfully completed when the iterative process has converged [9].

### 4.2. Modeling Performance Criteria

The accuracy and error quantification of the proposed system was evaluated using performance parameters. The first parameter is the R^2^ coefficient (coefficient of determination), which is the absolute fraction of variance of a variable. It is a measure of the proportion of the information in the data that is explained by the model [62]. The value of R^2^ varies from 0 to 1. The closer R² is to 1, the closer the forecast value is to the experimental one, expressed as (Equation (5)):(5)$$R^{2}=1-\left(\frac{\sum_{i=1}^{N}{\left(a_{PREDICT}-a_{TARGET}\right)}^{2}}{\sum_{i=1}^{N}{\left(a_{PREDICT}\right)}^{2}}\right)$$

The root mean square error (RMSE) is the square root of the mean square error and indicates the average distance of a data point (targeted) from the expected value (predicted) provided by the model. The lower the RMSE value, the better the model (Equation (6)):(6)$$RMSE=\sqrt{\left(\frac{1}{N}\right)*\left(\sum_{i=1}^{N}{\left(a_{PREDICT}-a_{TARGET}\right)}^{2}\right)}$$

For a better understanding, RMSE can be normalized using the mean of the actual value. This can facilitate comparisons between datasets or models [20].

MAPE (Equation (7)) is the mean absolute percentage error and is a statistical value of prediction accuracy. It indicates a better model fit through a percentage value. However, MAPE places a heavier penalty on negative errors than positive errors due to the division by the factor *a_PREDICT_*.
(7)$$MAPE=\left(\frac{100}{N}\right)*\left(\sum_{i=1}^{N}\left(\frac{\left|a_{PREDICT}-a_{TARGET}\right|}{\left|a_{PREDICT}\right|}\right)\right)$$

MAE is the mean absolute error formula and is given by Equation (8):(8)$$MAE=\left(\frac{1}{N}\right)*\left(\sum_{i=1}^{N}\left(\left|a_{PREDICT}-a_{TARGET}\right|\right)\right)$$

In all the formulae above:*N* is the number of experiments,*a_PREDICT_* = *a*
_(*j,*_*_PREDICT_*_)_ is the predicted value for the jth neuron*a_TARGET_* = *a _(j, TARGET_*_)_ is the experimental value for the jth neuron.

## 5. Bibliographic Dataset and Data Preparation

The comprehensiveness, structure, and volume of the data used for training are vital to building an effective network. This is what must lead to better learning, testing, and validating for the network and accurate prediction of all aspects of the relationship between inputs and outputs [20].

Experimental datasets from different sources were used. The notation used is given in Table 2. This is an inhomogeneous collection from the experimental data of some previous research work. The present database was built from the literature and includes a total of 1303 concrete formulations from 22 different studies. We used a large dataset to minimize the lack of data that causes informational uncertainty and to minimize model accuracy problems.

Data were assembled from the bibliography and 18 selected inputs were considered: the content of water, cement, and fine and coarse aggregate; admixture; age; and water/binder ratio; superplasticizer; the slump, etc. One output was considered, which is compressive strength.

This study takes into account a very wide spectrum of materials and quantities, as shown in Table 3. In Table 4**,** we also present an excerpt of concrete mixes from the dataset used. Part of the extensive list of formulations used for the ANN training - testing is given in the Appendix A.

## 6. Results and Discussion

The structure of the ANN applied in this study is shown in Figure 5 and Figure 6. The network consists of 18 inputs, two hidden layers, and one output, and was used for 1310 data values.

The results in Figure 7 show the model performance results measured through error minimization techniques. During the learning process, the error drops as the network is continuously trained. The patterns in Figure 7 are respectively training, validation, and testing relative to model error.

Pattern 1 (blue, Training) describes the training error obtained from 70% of the samples and improves the model’s fit by adjusting the network according to its error.Pattern 2 (green, Validation) fits the network generalization ability that instructed the network on when to stop the training process. Pattern 2 represents the ability of the model to predict new data [32] (predictive performance). The training process is halted when validation error stops decreasing, which inherently avoids over-fitting.Pattern 3 (red, Testing) does not affect training and is an independent measure of network performance. This error measured on the test data indicates how well the model is generalized to the data during and after training.

In Figure 7 and Figure 8, the results clearly demonstrate that the gradient begins to stabilize when the epoch equals 6. The values of the coefficient of determination R² are 0.9982, 0.9763, and 0.9566 for training, validation, and testing respectively. The histogram shown in Figure 9 shows that the error value is lowering from training to testing. This corresponds to an improvement of the model throughout the processing and precision. The high values of R² in Figure 10 mean that the model seems to have sufficient accuracy and is also well trained. 

In Figure 10, the network outputs with respect to targets for training, validation, and test sets are plotted against the target values. For a perfect fit, the data should fall along a 45° line, where the network outputs are equal to the targets. For this problem, the fit is very good for all datasets, with R² values of 0.95 or above in each case.

The performance indicators for compressive strength accuracy are given in Table 5. The value of RMSE, which is 2.91 MPa, shows that the gap between predicted and experimental values is small. MAPE shows that the predicted compressive strength deviated on average by 2.87% from the experimental data. This indicates that the differences between forecast and actual results were negligible. All these points indicate that the ANN strength predictive model was able to reproduce the experimental compressive strength results with high accuracy. These results are comparable to those obtained in similar studies [29,51]. In the same order, the determination coefficients reached by [39,78] are between 0.9443 and 0.9836.

## 7. Conclusions

This study aimed to use an artificial neural network to predict the compressive strength of waste-based concretes. The methodology, architecture, and learning methods were explained, based on feedforward and backpropagation techniques. A bibliographic dataset compiled from the literature was then used, including a total of 1303 concrete formulations from 22 different studies. The important conclusions that can be drawn from this work are:The ANN model can predict compressive strength with high accuracy by learning the deep features of the water–cement ratio, the cement and admixture content, the age of the concrete, etc.The results have demonstrated that multilayer feedforward artificial neural networks are practicable methods to forecast compressive strength in concretes.Errors of the model calculated from R², MSE, MAPE and MAE show small gaps between experimental and forecast values.

The above results suggest that the use of ANN is suitable for concrete compressive strength prediction. A coming study that we are undertaking will test the use of Decision Tree Regression (DTR) in the prediction of concrete properties. This machine learning method has been stated to be a very efficient approach.

## Figures and Tables

**Figure 1 materials-15-07045-f001:**
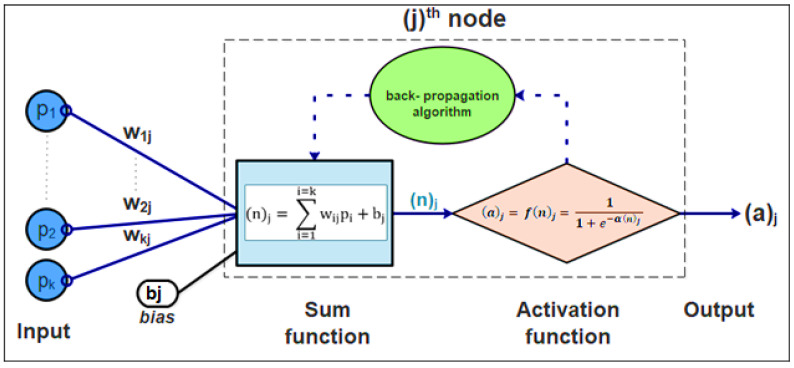
Architecture of artificial node and its interactions in the neural network.

**Figure 2 materials-15-07045-f002:**
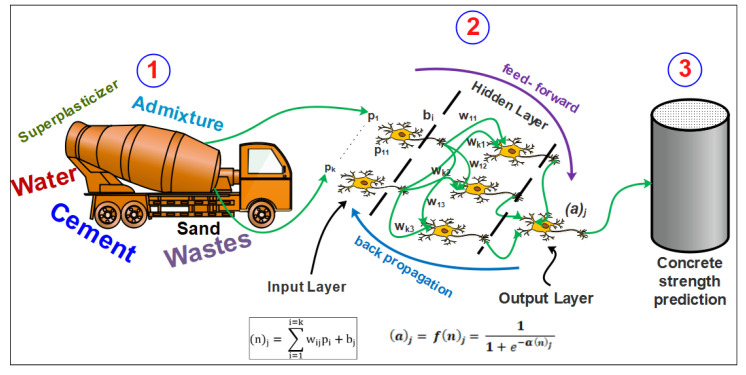
General scheme of the use of ANN for concrete design.

**Figure 3 materials-15-07045-f003:**
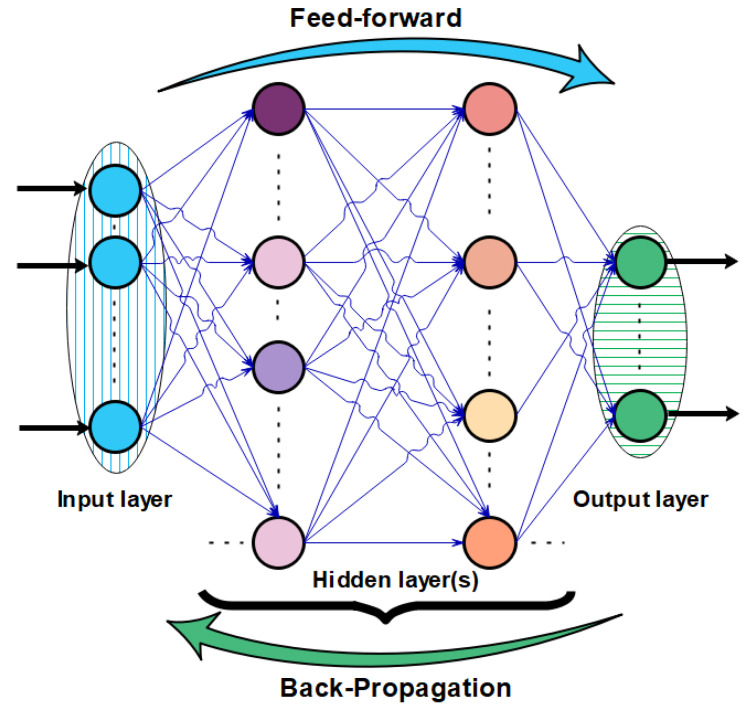
Architecture of applied neural network and learning process principle.

**Figure 4 materials-15-07045-f004:**
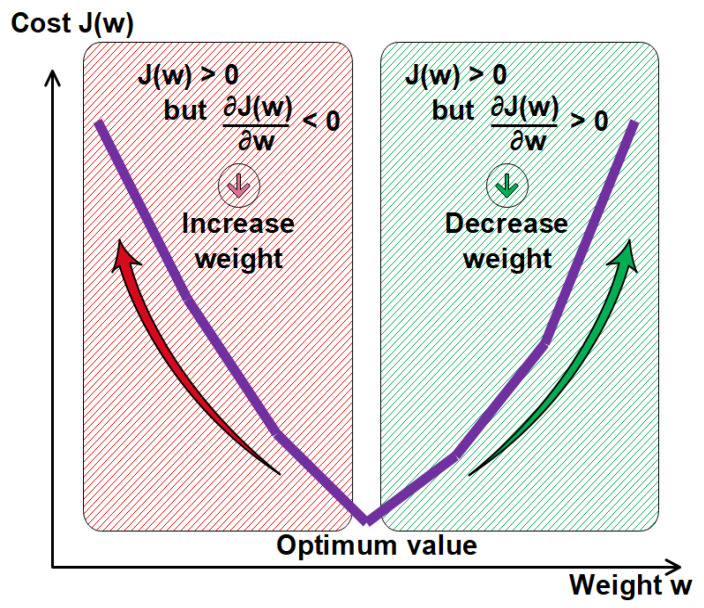
Relationship between cost and weight of the ANN.

**Figure 5 materials-15-07045-f005:**
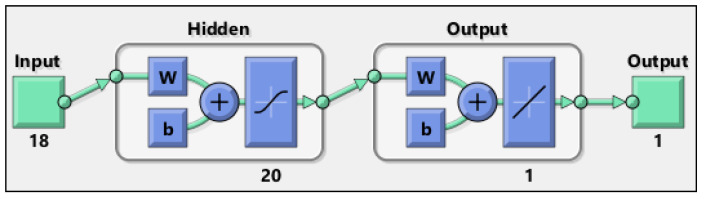
Artificial neural network architecture from MATLAB software (R2022a).

**Figure 6 materials-15-07045-f006:**
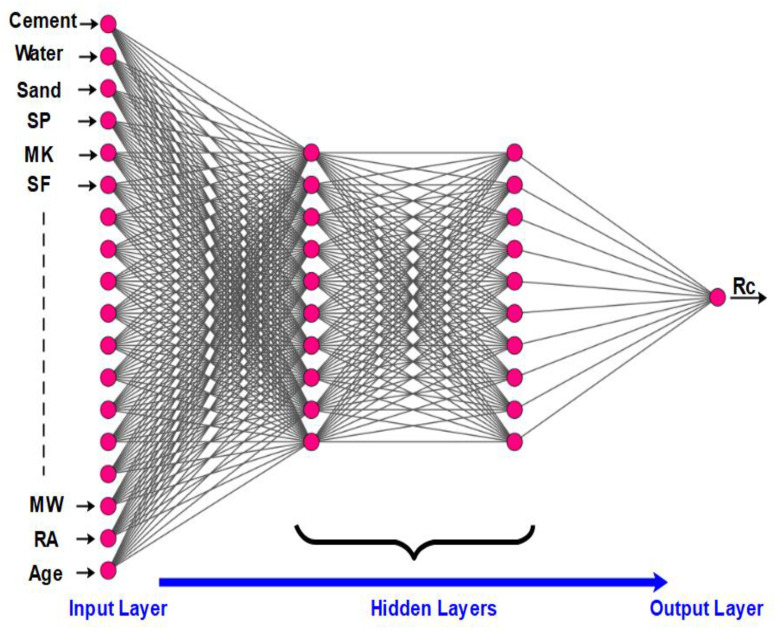
Architecture of the artificial neural network in this study.

**Figure 7 materials-15-07045-f007:**
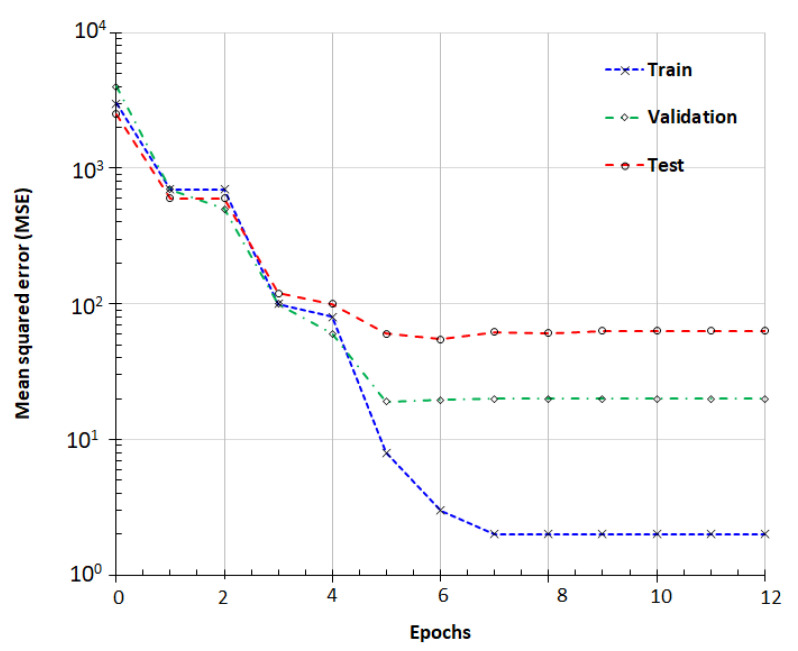
Model performance plot based on mean squared error (MSE) analysis.

**Figure 8 materials-15-07045-f008:**
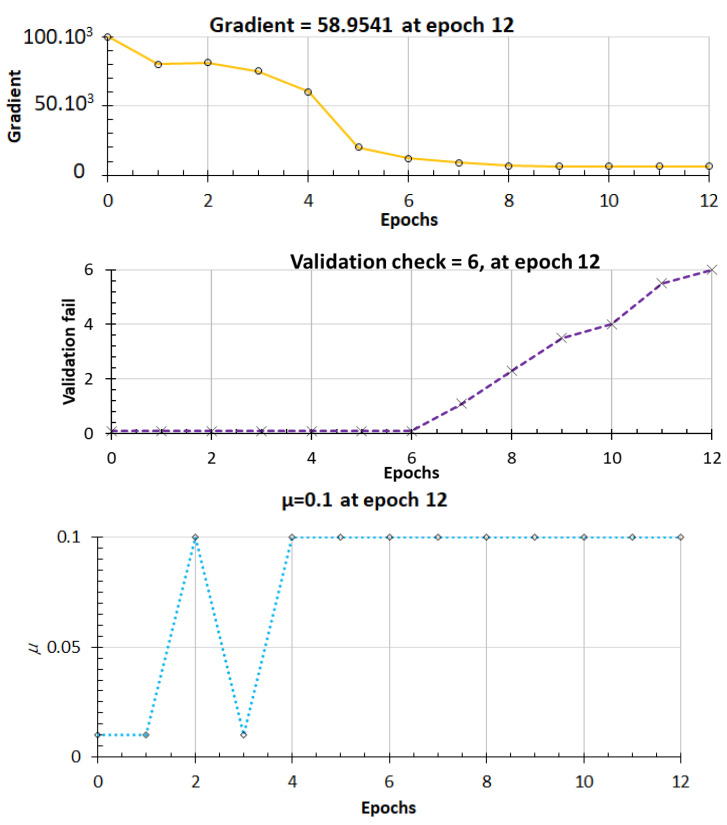
Plot of model performance evolution regarding the number of epochs.

**Figure 9 materials-15-07045-f009:**
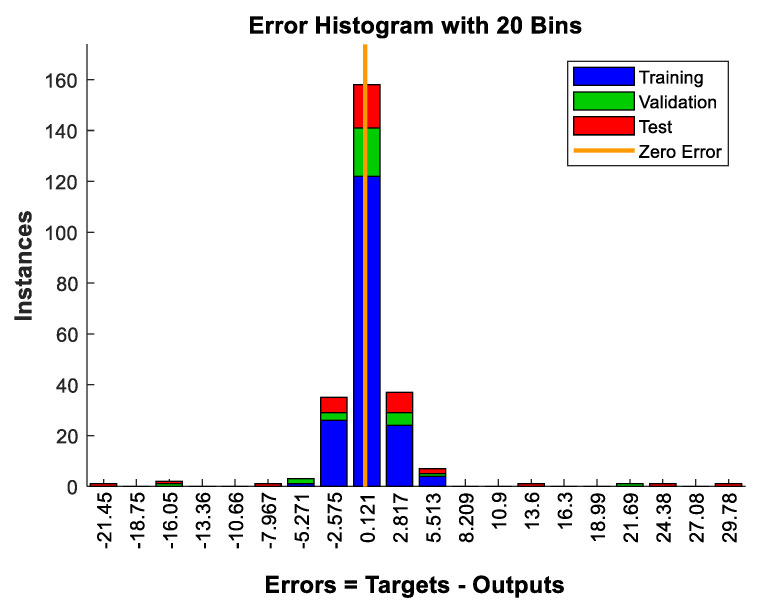
Training state results for performance checking.

**Figure 10 materials-15-07045-f010:**
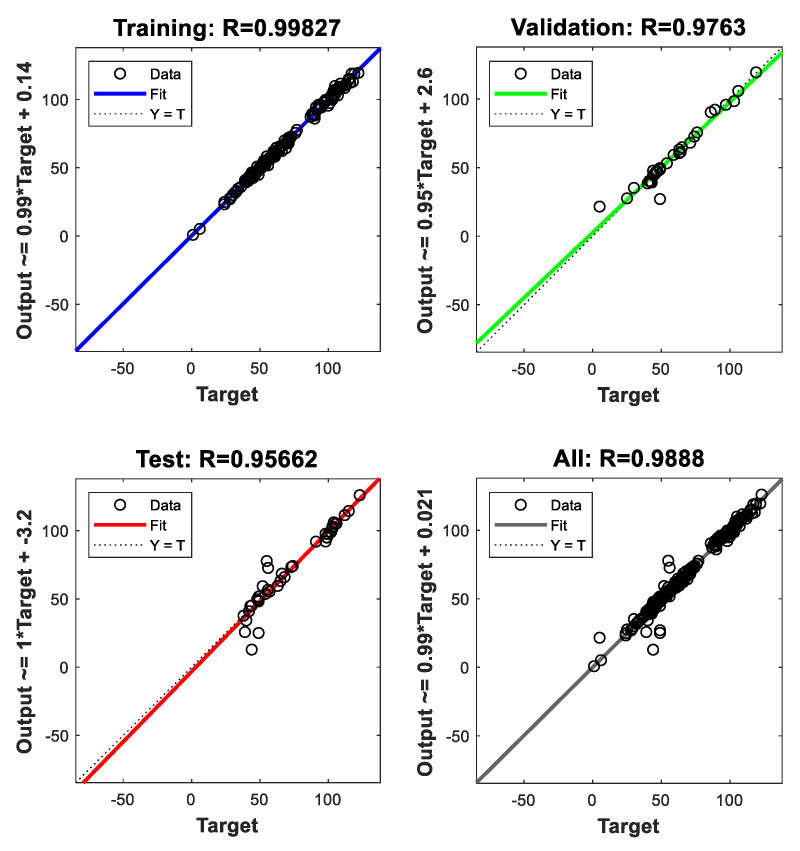
Regression analysis results.

**Table 1 materials-15-07045-t001:** General characteristics of the ANN model.

Method Type	Regulation Technique	Learning Method	Training Method	Activation Function	Inputs	Hidden Layers (HL)	Neurons per HL	Outputs
Feedforward, back propagation network	Gradient descent	Supervised	Levenberg-Marquardt backpropagation algorithm (LMBPA)	Log-sigmoid (logsig)	18	2	10	1

**Table 2 materials-15-07045-t002:** Data notation and abbreviations.

C	W	W/B	Specimen Compression Type	S	CA	SP	
Cement	Water	Water/Binder	1 = cubic 2 = cylindrical	Sand	Coarse aggregates	Superplasticizer	
MK	LF	SF	GGBFS	FA	MW	RA	Rc (MPa)
Metakaolin	Limestone filler	Silica fume	Ground granulated blast furnace slag	Fly ash	Marble waste	Recycled aggregates	Compressive strength

**Table 3 materials-15-07045-t003:** General characteristics of concrete formulations used in the dataset.

Parameter	C	E	MK	LF	SF	GGBFS	FA	MW	RA	W/B	Age (days)	Rc (MPa)
Minimum (Kg/m^3^)	70.0	95.0	0.0	0.0	0.0	0.0	0.0	0.0	0.0	0.08	1.0	1.0
Maximum (Kg/m^3^)	833.3	319.0	104.1	317.0	208.3	360.0	544.0	500.0	1772.0	0.72	365.0	123.0

**Table 4 materials-15-07045-t004:** Excerpt of dataset used in AI training and validation processes.

Mix N°	Author	C (Kg/m^3^)	W (Kg/m^3^)	W/B	Specimen Compression Type	S (Kg/m^3^)	Coarse Aggregates (Kg/m^3^)	SP (%)	Slump/Flow (mm)	MK (Kg/m^3^)	LF (Kg/m^3^)	SF (Kg/m^3^)	GGBFS (Kg/m^3^)	FA (Kg/m^3^)	MW (Kg/m^3^)	RA (Kg/m^3^)	Age (days)	Rc (MPa)
	[74]																	
1		280.0	202.0	0.72	2	777.0	988.0	0.0	160.0	0.0	0.0	0.0	0.0	0.0	0.0	0.0	1	5.3
2		224.0	185.0	0.66	2	788.0	1003.0	0.2	170.0	0.0	0.0	0.0	0.0	56.0	0.0	0.0	1	5.0
3		168.0	157.0	0.56	2	802.0	1041.0	0.8	180.0	0.0	0.0	0.0	0.0	112.0	0.0	0.0	1	3.9
4		112.0	124.0	0.44	2	801.0	1106.0	1.4	210.0	0.0	0.0	0.0	0.0	168.0	0.0	0.0	1	2.6
5		112.0	150.0	0.27	2	418.0	1101.0	0.7	220.0	0.0	0.0	0.0	0.0	448.0	0.0	0.0	1	2.3
6		340.0	203.0	0.60	2	737.0	977.0	0.1	220.0	0.0	0.0	0.0	0.0	0.0	0.0	0.0	1	7.6
7		272.0	188.0	0.55	2	743.0	985.0	0.2	210.0	0.0	0.0	0.0	0.0	68.0	0.0	0.0	1	7.6
	[10]																	
8		350.0	95.2	0.27	1	575.9	1273.0	0.0	---	0.0	0.0	0.0	0.0	0.0	0.0	0.0	28	61.1
9		350.0	98.5	0.28	1	558.2	1325.4	0.0	---	0.0	0.0	0.0	0.0	0.0	0.0	0.0	28	54.0
10		339.5	97.7	0.28	1	655.3	1273.0	0.0	---	0.0	10.5	0.0	0.0	0.0	0.0	0.0	28	65.7
11		339.5	97.6	0.28	1	535.0	1247.0	0.0	---	0.0	10.5	0.0	0.0	0.0	0.0	0.0	28	62.2
12		336.0	97.6	0.28	1	535.0	1247.0	0.0	---	0.0	14.0	0.0	0.0	0.0	0.0	0.0	28	54.5
13		332.5	97.7	0.28	1	655.3	1273.0	0.0	---	0.0	17.5	0.0	0.0	0.0	0.0	0.0	28	63.1
14		329.0	97.6	0.28	1	535.0	1247.0	0.0	---	0.0	21.0	0.0	0.0	0.0	0.0	0.0	28	52.2
	[75]																	
15		350.2	157.60	0.45	1	810.4	1200.6	0.0	5.0	0.0	0.0	0.0	0.0	0.0	0.0	0.0	1	19.07
16		332.2	157.30	0.45	1	809.2	1198.9	0.6	10.0	17.5	0.0	0.0	0.0	0.0	0.0	0.0	1	21.50
17		314.2	157.10	0.45	1	808.0	1197.0	1.2	15.0	34.9	0.0	0.0	0.0	0.0	0.0	0.0	1	22.43
18		296.3	156.90	0.45	1	806.8	1195.3	1.8	25.0	52.3	0.0	0.0	0.0	0.0	0.0	0.0	1	20.23
19		278.5	156.70	0.45	1	805.6	1193.6	2.4	75.0	69.6	0.0	0.0	0.0	0.0	0.0	0.0	1	19.33
20		260.7	156.40	0.45	1	804.5	1191.8	3.0	75.0	86.9	0.0	0.0	0.0	0.0	0.0	0.0	1	15.73
21		243.0	156.20	0.45	1	803.3	1190.0	3.6	90.0	104.1	0.0	0.0	0.0	0.0	0.0	0.0	1	14.53
22		350.2	157.60	0.45	1	810.4	1200.6	0.0	5.0	0.0	0.0	0.0	0.0	0.0	0.0	0.0	7	50.23
23		332.2	157.30	0.45	1	809.2	1198.9	0.6	10.0	17.5	0.0	0.0	0.0	0.0	0.0	0.0	7	53.80
	[76]																	
24		300.0	165.0	0.41	1	1095.0	722.0	1.0	30.0	0.0	0.0	0.0	0.0	0.0	100.0	0.0	7	25.8
25		300.0	165.0	0.41	1	1095.0	722.0	2.0	57.0	0.0	0.0	0.0	0.0	0.0	100.0	0.0	7	30.7
26		300.0	165.0	0.41	1	1095.0	722.0	3.0	58.0	0.0	0.0	0.0	0.0	0.0	100.0	0.0	7	22.2
27		300.0	180.0	0.45	1	1071.0	706.0	1.0	43.0	0.0	0.0	0.0	0.0	0.0	100.0	0.0	7	25.8
28		300.0	180.0	0.45	1	1071.0	706.0	2.0	60.0	0.0	0.0	0.0	0.0	0.0	100.0	0.0	7	28.9
29		300.0	189.0	0.47	1	1055.0	696.0	2.0	66.0	0.0	0.0	0.0	0.0	0.0	100.0	0.0	7	27.6
30		300.0	201.0	0.50	1	1039.0	685.0	2.0	68.0	0.0	0.0	0.0	0.0	0.0	100.0	0.0	7	26.2
	[77]																	
38		553.5	161.6	0.10	2	734.2	0.0	0.0	0.0	0.0	0.0	0.0	0.0	0.0	0.0	1145.9	7	51.6
39		524.5	205.5	0.13	2	695.7	0.0	0.0	55.0	0.0	0.0	0.0	0.0	0.0	0.0	1085.8	7	40.4
40		498.3	245.2	0.16	2	661.0	0.0	0.0	179.0	0.0	0.0	0.0	0.0	0.0	0.0	1031.6	7	28.9
41		474.7	280.9	0.19	2	629.7	0.0	0.0	531.0	0.0	0.0	0.0	0.0	0.0	0.0	982.7	7	24.6
42		553.5	152.0	0.08	2	734.2	0.0	0.0	0.0	0.0	0.0	0.0	0.0	0.0	0.0	1351.5	7	62.1
43		524.5	196.4	0.11	2	695.7	0.0	0.0	32.0	0.0	0.0	0.0	0.0	0.0	0.0	1280.6	7	46.4
44		498.3	229.4	0.13	2	661.0	0.0	0.0	180.0	0.0	0.0	0.0	0.0	0.0	0.0	1216.7	7	33.6
45		474.7	272.7	0.17	2	629.7	0.0	0.0	563.0	0.0	0.0	0.0	0.0	0.0	0.0	1159.1	7	27.3

**Table 5 materials-15-07045-t005:** Values of model performance accuracy parameters.

Parameters	Performance Indicators				
	RMSE (MPa)	R²	MSE (MPa)	MAE (MPa)	MAPE (%)
**Values**	2.91	0.9888	8.4689	1.7463	2.87

## Data Availability

The data used in this study are available from the corresponding author on submission of a reasonable request.

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
