# Peer review of "Prediction of the Compressive Strength of Waste-Based Concretes Using Artificial Neural Network"

_materials, 2022, doi:10.3390/ma15207045_

Round 1
Reviewer 1 Report
1 What is the size of cubic and cylindrical specimens in table 2? Do they have the same size in all references?
2 The numbering of sections is wrong. For example, "1.1. Neuron Model (logsig, tansig, purelin)" follows section 3. "1.4. The backpropagation algorithm (BPA)" follows "4. The learning and testing process".
3 Please provide an appendix of the data of 1303 specimens. I only see 45 specimens in the manuscript.
4 What is the Superplasticizer used in 1303 specimens? Are they the same?
5 Do MK, LF and SF used in 1303 specimens have the same size?
6 Will the compressive strength be beyond 123 MPa? UHPC often has a compressive strength greater than 123 MPa.
7 Conclusions have lost the numbering.
8 There are many different methods used in the prediction of concrete, like SVM [1] and Regression Tree [2]. Please cite the reference and make a comparison with ANN.
Ref.:
[1] Machine learning models for predicting the compressive strength of concrete containing nano silica. Computers and concrete, 2022, 30, 1, 33-42.
[2] The Prediction of Self-Healing Capacity of Bacteria-Based Concrete Using Machine Learning Approaches. COMPUTERS MATERIALS & CONTINUA, 2019, 1, 57-77.
9 ANN has been widely used in concrete [3-4]. Please cite the reference and complete the introduction.
Ref.:
[3] PREDICTION OF EARLY COMPRESSIVE STRENGTH OF ULTRAHIGHPERFORMANCE CONCRETE USING MACHINE LEARNING METHODS. International Journal of Computational Methods, 2022, Early Access, https://doi.org/10.1142/S0219876221410231
[4] Experimental investigation and comparative machine-learning prediction of compressive strength of recycled aggregate concrete. Soft computing, 2021, 25 (2), 919-932
Reviewer 2 Report
Amar et al. report the usage of an artificial neural network (ANN) to forecast the compressive strength of waste-based concretes. The article is well organized and the work is very interesting. Therefore, I suggest that the paper can be accepted after minor revision. Some suggestions are as follows:
1. It is mentioned in the abstract that the model has high prediction accuracy. Please provide specific values in the abstract.
2. Please compare the prediction accuracy of your results with other literature models to reflect the model accuracy.
3. What is the innovation of this manuscript? Please add to the manuscript.
4. Please correct any minor errors in format, sentence tenses, units of physical quantities etc. of your manuscript.
5. Reference selection [17], [86-89], what is the representativeness of the experimental data cited? Why choose these papers? Authors should be aware that the cited reference paper is published too early.
Reviewer 3 Report
Dear colleagues,
The article is interesting and well-written. However, it requires correction before publication. Below are some comments that will help in the final form of the article.
General comments
- The manuscript is clear, relevant for the field, and presented in a well-structured manner.
- The cited references are relevant.
- The manuscript is scientifically sound, the experiments are appropriately designed, and the methods are well described.
- The figures and tables are appropriate. They properly show the data and are easy to interpret and understand.
Due to the similarity in the presentation of the results in Figures 7-10 to those presented in the article "Prediction of compressive strength of recycled aggregate concrete using artificial neural network and cuckoo search method" by Catherina Vasanthalin P. and Chella Kavitha N., I suggest to change the artwork of the figures. - According to the article and numerous publications, artificial intelligence methods (including artificial neural networks) for predicting concrete properties have been successfully used for many years. Therefore, the authors should clearly outline the novelty of their research.
Expanding the concluding chapter with conclusions based on new solutions used in the presented analysis can significantly increase the article's value.
Specific comments
The manuscript requires minor linguistic and editorial corrections, for example:
Title - “an” article before “artificial neural network”
Line 25 - “worldwide” instead of “in the world”
Line 51 - globally instead of “in the world”
Line 119 - unnecessary dot
Table 4 - the titles of individual columns need to be improved
Line 369 – Fig. 10 instead of Fig. 1
Reviewer 4 Report
Dear authors,
similarly to your approach I understand machine learning as an interesting tool for accelerating some of the engineering tasks, especially if e.g. composite material behavior heeds to be predicted faster reducing the demand for time-consuming testing. In general the manuscript is well writtne, incl. nice introduction with clear summary of various approaches done in machine learning or deep learning. I have only two or three comments:
Table 3: the parameters "E" and E/L" are not explained in the text (maybe I have overlooked them).
The trainign results and nalysis are very good and supporting the fact that ANN can be used for predicting copressive strength of concrete based on some input values. Nevertheless I would be interested to see some practical applicaiton on soe examples for concrete mixtures you maybe have envisaged to design using e.g. 50% waste based concrete. Additionally I voudl be interested if you have realized any risk if waste (recycled) concrete will be used e.g. as part of typical coarse aggregates since usually the recycled aggregates have a it different properties and usuaylly a bit more heterogeneity. Is there any impact of these aspects on the learnibng process and its accuracy?
Reviewer 5 Report
The authors took up an interesting topic worth exploring and research. However, in my opinion, the manuscript overrepresents knowledge that is generally available elsewhere (basics and principles of neural networks, error functions, etc.), and the essence of the research itself has been treated quite superficially. It would be much more interesting to investigate the influence of network architecture or alternative methods of learning and error propagation on the obtained results. Perhaps then it would be possible to compare and optimize variants of solutions. In my opinion, the publication of an article must be preceded by such work.
Minor comments: no explanation of symbols (E and E / L) is given in Table 3.
Round 2
Reviewer 1 Report
The manuscript has been revised properly.
Reviewer 5 Report
In my opinion the current version of the manuscript can be accepted.